# Occidiofungin: Actin Binding as a Novel Mechanism of Action in an Antifungal Agent

**DOI:** 10.3390/antibiotics11091143

**Published:** 2022-08-23

**Authors:** Nopakorn Hansanant, Leif Smith

**Affiliations:** Department of Biology, Texas A&M University, College Station, TX 77843, USA

**Keywords:** natural products, drug discovery, antimicrobial targets, antifungal, mechanism of action

## Abstract

The identification and development of natural products into novel antimicrobial agents is crucial to combat the rise of multidrug-resistant microorganisms. Clinical fungal isolates have been identified, which have shown resistance to all current clinical antifungals, highlighting a significant need to develop a novel antifungal agent. One of the natural products produced by the bacterium *Burkholderia contaminans* MS14 is the glycolipopeptide occidiofungin. Occidiofungin has demonstrated in vitro activity against a multitude of fungal species, including multidrug-resistant *Candida auris* strains, and in vivo effectiveness in treating vulvovaginal candidiasis. Characterization of occidiofungin revealed the mechanism of action as binding to actin to disrupt higher-order actin-mediated functions leading to the induction of apoptosis in fungal cells. Occidiofungin is the first small molecule capable of disrupting higher-order actin functions and is a first-in-class compound that is able to circumvent current antifungal resistant mechanisms by fungal species. Anticancer properties and antiparasitic activities, against *Cryptosporidium parvum*, have also been demonstrated in vitro. The novel mechanism of action and wide spectrum of activity highlights the potential of occidiofungin to be developed for clinical use.

## 1. Introduction

Fungal infections constitute a large financial and medical burden globally, with over a million deaths associated with fungal infections globally every year [1]. Morbidity and mortality are exacerbated by the rise of drug-resistance fungal species and strains as well as the narrow repertoire of clinically available antifungal agents. *Candida* species constitutes the majority of fungal infections, and resistance has been seen to first-line drugs fluconazole and echinocandins [2]. *C. auris* strains have been isolated, which show resistance to all clinically available antifungal agents [3]. This is particularly concerning in community settings such as hospitals or senior-assisted living facilities where outbreaks are particularly dangerous for the susceptible population. Drug resistance is also seen in other non-candida species such as *Aspergillus fumigatus*, *Cryptococcus neoformans,* and *Pneumocytis jirovecii* [4,5,6].

There is an urgent need to develop new antifungal agents; however, there has been no new clinically approved class of antifungals for the treatment of serious fungal infections since the echinocandins in the early 2000s. Current classes of antifungal agents are limited in the spectrum of their mechanism of action, as they either inhibit ergosterol biosynthesis (azoles and polyenes) or Beta-glucan biosynthesis (echinocandins). Resistance to drugs in these three classes has been seen in various clinical fungal isolates, including multidrug-resistant strains. Amphotericin B is considered a last-line treatment option, although resistant strains have emerged as a result of treatment [7]. The natural product occidiofungin differs from other antifungal agents as it induces cellular apoptosis in fungal cells [8]. The cellular target is actin, but unlike other small actin-binding compounds characterized to date, occidiofungin does not inhibit in vitro actin polymerization or depolymerization. Instead, occidiofungin interferes with higher-order actin cable function [9], which triggers a cascade to signal the induction of apoptosis [10]. Mitochondria and actin interaction are vital for mitochondrial functions. Actin has also been linked to the regulation of mitochondrial permeability, mitochondria homeostasis, and the concentration of mitochondria to areas with high energy demands. Mitochondria has been shown to be able to induce cellular apoptosis, and disruption of actin–mitochondria interaction could also induce apoptosis [10]. Actin also plays a key role in signaling mechanisms as a result of environmental nutrition, which can induce apoptosis [10]. No clinically relevant fungal species has yet been identified that is resistant to occidiofungin, and this highlights the possibility for occidiofungin to be developed as an antifungal agent.

## 2. Occidiofungin Characterization

The bacterium *B. contaminans* belongs to the *Burkholderia cepacia* complex (Bcc) group of bacteria. These bacteria are found in the natural environment such as in soil; however, they have been isolated in clinical settings particularly in immunocompromised individuals. Bcc species are noted for their secondary metabolites for antimicrobial properties and quorum sensing. *B. contaminans* MS14 strain was first observed to have antifungal properties in 2009, which were later attributed to occidiofungin [11]. Occidiofungin is a hybrid polyketide and non-ribosomally synthesized peptide consisting of eight amino acids (Figure 1).

Bcc species have been characterized to produce a variety of antifungal compounds including burkholdines, cepacidines, and xylocandins [13]. There is a high degree of similarity in the chemical structure of these compounds and with occidiofungin. They are all cyclic lipopeptides, and most variants of these compounds have an attached xylose sugar at the side chain of the lipid group [13]. The differences in each antifungal produced may be a result of the fungal species encountered in the natural environment of each Bcc species. Although occidiofungin has been shown to have broad-range in vitro antifungal activity, the unique structure may be more catered towards fungi found in the natural environment of *B. contaminans* MS14.

The biosynthesis of occidiofungin in *B. contaminans* MS14 was identified to be a 58.1-kB gene cluster named the *ocf* gene cluster, *ocfA* to *ocfN* (Table 1) [12]. The *ocf* gene cluster shares a high degree of similarity with an uncharacterized DNA region of *Burkholderia ambifara* strain AMMD, and the flanking regions of the gene cluster showed similar G+C content as their homologs in *Burkholderia lata 383*. The *ocf* gene cluster may have been horizontally transferred from a *B. ambifara* AMMD and integrated into *B. lata* 383, which diverged into *B. contaminans* MS14. Genomic comparison between *B. contaminans* MS14 and *B. lata* 383 shows high similarity, although *B. lata 383* is missing *ocfD*-*ocfJ* homologs, and these two species were previously classified as taxon K of the Bcc species. Occidiofungin has also been identified to be produced by other *B. contaminans* strains, such as MS455 isolated from soybean, which showed broad-range antifungal activity including against *Aspergillus flavus* [14]. No other similar gene clusters are found in other *Burkholderia* species including those that have been identified to be pathogens in clinical cases. Distinguishing between pathogenic *Burkholderia* species and those that are less virulent is important for biological applications, such as plant growth promotion or antimicrobial production. The absence of the *ocf* gene cluster and potential homologs in characterized pathogenic *Burkholderia* species suggests that this gene cluster is not involved in pathogenicity [15].

Expression of the gene cluster is under the control of two LuxR-type regulators ambR1 and ambR2 (Table 1) [16]. Knockout of ambR1 resulted in loss of antifungal activity, and knockout of ambR2 showed reduced antifungal activity. Expression of ambR1 regulated the transcription of ambR2, and constitutive expression of ambR2 in the absence of ambR1 does not compensate for the loss of ambR1. The importance of ambR1 is greater than ambR2 in occidiofungin production, which may be due to ambR1 encoding a transcription regulator for the entire *ocf* gene cluster.

Non-ribosomal peptide (NRP) synthetases can incorporate non-conventional amino acids during non-ribosomal peptide synthesis (NRPS). The recognition and incorporation of these amino acids is important for the function of the final peptide. Novel amino acid 2 (NAA2) is synthesized in a polyketide pathway where it is then incorporated into the growing peptide chain in the NRPS pathway (Figure 2) [12]. The promiscuous nature of non-ribosomal peptide synthesis can lead to peptide variants due to the incorporation of non-specific amino acids [17,18]. For occidiofungin, the first step in the NRPS pathway can incorporate either asparagine (ASN1) or beta-hydroxy-asparagine (BHN1) by OcfJ. Conformational variants have also been shown to play a role in the antifungal activity of occidiofungin. The final step in the NRPS pathway involves cyclization and release of the peptide mediated by thioesterase proteins. The *ocf* gene cluster encodes for two thioesterases, OcfD and OcfN, and both are important for producing conformational variants of occidiofungin. OcfD thioesterase module is located at the C-terminal end of the OcfD NRP synthetase, whereas OcfN is an independently expressed thioesterase gene product. Knockout of OcfN resulted in decreased production of ASN1 occidiofungin and decreased antifungal activity [19]. In the absence of OcfN, OcfD is still able to recognize ASN1 occidiofungin but fewer stereoisomers of ASN1 occidiofungin are present compared to wild-type MS14. For *B. contaminans* MS14 to evolve and express two thioesterases, the variety of conformational variants of occidiofungin must be important in antifungal activity. Each variant may be more suited for different fungal species the bacteria encounters in the natural environment.

The beta-hydroxy-tyrosine 4 (BHY4) has also been observed to be chlorinated by the halogenase OcfK. It is unknown whether NRP synthetase OcfG is selective in recognition of BHY4 and OcfK chlorinates BHY4 after recognition, or whether OcfG can recognize both BHY4 and chlorinated BHY4. *B. contaminans* MS14 may have evolved the halogenase as a mechanism to increase the diversity of occidiofungin variants to maximize antifungal activity in their natural environments. The incorporation of the chlorine may also be a way for the bacteria to excrete toxic chlorine-based substances as a survival mechanism.

Occidiofungin has a xylose sugar on the side chain of NAA2, attached by the glycosyltransferase OcfC. Knockout of *ocfC* resulted in xylose-free occidiofungin, which had similar in vitro antifungal activity as occidiofungin, but greatly decreased yield [20]. The presence of the xylose sugar may be important for secretion or in the solubility of occidiofungin. It is unlikely that xylose is needed for recognition of NAA2 by OcfH and OcfJ in the NRPS pathway since there is still production of xylose-free occidiofungin even with disruption of *ocfC*. Further studies on the role and importance of xylose for occidiofungin would be important since xylose is not naturally found in mammals. A mild inflammatory response is observed in repeat-dose toxicity studies in mice, which may be due to the xylose.

## 3. Bioactivity

Occidiofungin binds to actin to induce cellular apoptosis [8]. Actin-binding has been documented to be able to signal apoptosis in fungal cells [21]. Actin can spontaneously polymerize and depolymerize between the monomeric globular form and filamentous form. Actin filaments form higher-order structures in yeast cells such as actin patches, rings, and cables which facilitate crucial cellular functions such as endocytosis, hyphae induction and nuclear segregation [22]. Pharmaceutical compounds which bind to actin prevent polymerization or depolymerization and are used mainly for chemotherapeutic purposes. Since actin is crucial for cellular function, the effect of these drugs is often toxic in humans and many natural products, such as phalloidin, which bind to actin are unable to be used as pharmaceuticals. Occidiofungin has a more subtle effect on actin dynamic by disrupting higher-order actin structures rather than disruption of polymerization and depolymerization of actin filaments [9]. Occidiofungin has a lower effective toxic concentration resulting in 50% cell death (TC_50_) in fungal and cancer cell lines, compared to healthy human fibroblasts [23]. This indicates that there is a suitable treatment dosage that would be therapeutic without displaying overt toxicity. The minimum lethal concentration of occidiofungin in human serum is also not much higher compared to the minimum inhibitory concentration in vitro and is below the TC_50_ value for fibroblast cells [23,24].

Fluorescent microscopy of a semi-synthetic analog of occidiofungin, which had lower antifungal activity, showed localization where actin is documented to be within yeast cells (Figure 3) [9]. *Saccharomyces cerevisiae* and *Schizosaccharomyces pombe* were both tested. *S. cerevisiae* is a budding yeast and divides through the production of bud tips, whereas *S. pombe* is a fission yeast that divides through medial fission. Exposure for 10 min of the occidiofungin analog shows localization at the bud tips of *S. cerevisiae*. Exposure of 30 min showed greater fluorescent signal at the bud tips of *S. cerevisiae*, and generally throughout the cell at endocytic vesicles, and the cell tips and division septum of *S. pombe*. These locations are documented to be where actin is located during cellular division for these species. Occidiofungin had no effect on the polymerization or depolymerization of filamentous actin in vitro, suggesting that the actin binding induces a change in actin dynamics to disrupt actin-mediated functions and inducing apoptosis [9]. The disruption of higher-order actin structures is seen through the loss in integrity of actin cables and patches [9]. It may be possible that occidiofungin binding may affect disrupt the function of other actin-binding proteins vital for the formation or function of actin structures. The difference between occidiofungin and other actin-binding compounds highlights the potential for occidiofungin to be developed as an antifungal agent. Complete disruption of polymerization and depolymerization results in cellular toxicity, but disruption of higher-order actin structures may be more targeted to fungal cells than other eukaryotic cells.

Actin co-sedimentation assays revealed that occidiofungin had a dissociation constant (Kd) of 1050 nM and stoichiometry of 24:1 [9]. The high binding ratio of occidiofungin binding to actin suggests that the dissociation constant observed is macroscopic for the 24 occidiofungin molecules. The high binding ratio of occidiofungin to actin is reminiscent of an aggregate formed through self-assembly when a molecule has a hydrophobic “tail” and hydrophilic region. Self-assembly is a common feature of many lipopeptides and the side chain of NAA2 could act as the hydrophobic tail for occidiofungin to form micelle-like structures around actin. The exact mechanism by which occidiofungin binding to actin induces cellular apoptosis is unknown, but the high binding ratio likely plays an important role in the antifungal activity [25].

Occidiofungin shows sub-micromolar antifungal activity to a wide variety of fungal species, including multidrug-resistant *C. auris,* which is a growing concern in clinical and community environments (Table 2) [9]. Exposure of fungal cells to occidiofungin showed loss of endocytosis, loss of higher-order filamentous actin structures, and reduction in hyphal morphology in *C. albicans* [9]. An increase in double-stranded DNA breaks and reactive oxygen species is also observed. Oxidative stress can reduce the dynamic capabilities of actin, in switching from globular or filamentous form, leading to apoptosis. Deletion of the gene *yca1P* in *S. cerevisiae* increased the minimum inhibitory concentration value by twofold to occidiofungin [8]. Yca1P is a caspase-like enzyme that degrades proteins under oxidation conditions, which contribute to apoptosis cell death. The effectiveness of occidiofungin on disrupting fungal biofilm formation has not yet been determined. Actin plays a key role in biofilm formation, so it may be possible that occidiofungin could inhibit biofilm formation, and this assumption is supported by the in vivo efficacy of occidiofungin in treating a vulvovaginal yeast infection [26].

Occidiofungin has also been tested for anticancer activity in vitro since actin is a highly conserved protein amongst eukaryotic organisms. Currently, there are no clinical anticancer drugs that target actin. There are several compounds which target the polymerization and depolymerization of microtubules, which is another cytoskeletal protein. These are used as anticancer drugs such as Taxol^®^ and vinblastine, although these drugs are known to have moderate to severe side effects since they also act on normal human cells [27,28]. Occidiofungin anticancer activity was observed against ovarian, brain, and B-cell lymphoma cancer cell lines [23]. The cancer cells lines had toxicity concentration resulting in 50% cellular death (TC_50_), which was between 60–70 nM for the three cell lines tested. This was much lower compared to the TC_50_ of occidiofungin to normal human fibroblasts, which was 533 nM. Bortezomid had TC_50_ of 4091 nM against fibroblasts and is commonly used to as a treatment for myeloma and lymphoma in clinical settings. The lower TC_50_ value of occidiofungin indicates a higher potency compared to Bortezomid, and the range between TC_50_ to normal fibroblasts and cancer cells highlights a dosage level which may be effective without severe side effects. Occidiofungin could be the first actin-binding class of drugs used in anticancer treatment which has similar safety margins compared to the tubulin-binding drug class.

Antiparasitic activity has also been investigated in vitro against *Cryptosporidium parvum,* which is an intestinal protozoan parasite in humans [29]. *C. parvum* can be contracted through ingestion of contaminated water or fecal matter and fully effective treatment is not yet available. Occidiofungin was observed to be poorly absorbed in the intestinal tract of mice, and retention in this area may make it a prime candidate for treatment of *C. parvum*. Low-nanomolar antiparasitic activity was observed by occidiofungin, which was comparable to other known clinical compounds. The effective concentration where 50% of parasitic cells were killed (EC_50_) was 9–15 times lower than the TC_50_ for the cell lines tested, indicating potential as a therapeutic agent. Occidiofungin acted on the various stages of asexual reproduction of the parasite, including the motile and infectious sporozoites. The antiparasitic activity was also demonstrated to be non-reversible following removal of occidiofungin, which improves the treatment efficacy. Sporozoites treated with occidiofungin were shorter and more transparent, which may be due to disruption of the cytoplasmic membrane and loss of cytosolic content. Whether occidiofungin also acts on actin within the parasites is uncertain, although actin is known to be highly conserved between eukaryotic organisms, making it a probably target.

The effectiveness of occidiofungin to cancer and parasitic cells compared to healthy human cells has not yet been fully investigated. Actin is likely to be the target of occidiofungin in parasitic and cancer cells, although this has not been determined. Cancer cells are known to be more metabolically active compared to healthy mammalian cells [30]. Metabolic changes occur in cancer cells for nutrient acquisition, which is necessary to fulfill the cell’s energetic needs in unregulated growth. For example, glucose uptake and fermentation to lactate are significantly increased in cancer cells in a process called the Warburg effect [31]. The uptake of extracellular components by these more metabolically active cells may inadvertently increase the uptake of occidiofungin, which is not seen in healthy mammalian cells. Unicellular organisms such as C. parvum, as well as yeast cells, may be more susceptible to occidiofungin due to increased metabolic activities compared to healthy mammalian cells. Interestingly, yeast cells in a state of quiescence had short-term resistance to occidiofungin, which may be due to slow uptake of extracellular material before entering the cell cycle [32]. Yeast cells where cytosolic translation was inhibited also showed protection against occidiofungin, suggesting that the rate of protein synthesis is linked to the antifungal activity of occidiofungin [32]. Resistance to occidiofungin was also demonstrated in cultures supplemented with calcium chloride or magnesium chloride in a dose-dependent manner, although the exact mechanism of resistance has not been elucidated [32].

The chemical stability of occidiofungin has been tested under heat, pH, and protease treatment. In vitro antifungal activity was not affected by treatment of heating up to 100 °C for 30 min, incubating from pH 2–9 for 30 min, or treatment with gastric proteases trypsin, chymotrypsin, and pepsin for 120 min [24]. The cyclical structure of occidiofungin may confer protection against these chemical conditions. The chemical stability of occidiofungin is useful for development as a pharmaceutical agent by reducing degradation, increasing storage life, and allowing for different routes and methods of administration.

## 4. Clinical Studies

Occidiofungin has been tested extensively on mice for toxicity and efficacy [33]. Intraperitoneal (IP) injection up to 20 mg/kg showed no behavioral changes associated with severe pain. Body weight loss was observed to be dosage-dependent with a loss of up to 12% body weight at a single dosage of 20 mg/kg. Repeat intravenous-dose injection at 2 mg/kg per 48-h interval for 28 days showed similar weight changes as single-dose injection at 5 mg/kg. No weight loss of more than 20% was observed, indicating no severe toxicity in the mice. Recovery of body weight was observed in the days post injection. At 1 mg/kg dosage, which is like the treatment dosage used for echinocandins in humans, occidiofungin did not significantly affect body weight. The weight changes observed may be due to reduced intake of food rather than organ damages.

Serum chemistry and hematology analysis showed no statistically significant changes 5 days post injection of 10 mg/kg subcutaneously. In a 5-day repeat-dose toxicity study of 2 mg/kg given intraperitoneally, increased levels of neutrophils and decreased levels of lymphocytes was detected. This may be due to a non-specific mild inflammatory response to occidiofungin, which could be due to the xylose sugar found on the NAA2 side chain. However, no consistent dose-response effects were noted in both serum chemistry and hematology indicating that any observed results may be due to natural variations.

Decreased thymus weight was observed in the repeat-dose toxicity study, which is another sign of a non-specific inflammatory response. In a single dose of 10 mg/kg, liver weights were increased, although no histological changes were observed. No increase in alanine transaminase and aspartate aminotransferase was detected 5 days post dosage indicating no long-term liver damage. Overall, these findings suggest a mild liver response to occidiofungin. These observed results are seen in dosage levels that are higher than the typical 1 mg/kg of other antifungal drugs. The dosages used to determine toxicity is greater than levels of occidiofungin needed for antifungal activity in vitro. No significant weight changes were detected in other organs such as brain, kidney, spleen, lung, and thymus in both single- and repeat-dose studies. The relative weight percentages of organs in terms of body weight increased due to the general reduction in weight.

Vulvovaginal candidiasis (VVC) is a superficial infection that has not had new development in terms of therapeutics for decades. Occidiofungin was evaluated for activity in treating VVC in mice models, given that no significant toxicity was observed [9]. Mice treated with occidiofungin, in a gel formulation through suspension with noble agar, showed a significant 2-log reduction in *C. albicans* colony forming units compared to vehicle control. Treatment with 50 and 100 µg both showed 2-log reduction; however, no statistical significance was observed between the two dosages suggesting that a higher dosage may be able to further reduce fungal load. Treated mice showed no clinical changes associated with stress and no bleeding or swelling of the vagina was observed. Histological examination of the lung, kidney, heart, and spleen showed no significant *damage.* The mucus lining of the vaginal epithelium and the uppermost layer of cornified epithelial cells may confer protection to lower layers of cells and contribute to the lack of toxicity observed. A second VVC study using an occidiofungin gel formulation at 0.5%, 0.25%, and 0.1% demonstrated a dose-dependent response. Treatment with 0.1% occidiofungin showed no significant reduction in fungal load, 0.25% showed a 1-log reduction, and 0.5% showed 2-log rection. Monistat 3^TM^, a commercially available treatment for VVC, showed a 1-log reduction comparable to the 0.25% occidiofungin formulation. The effectiveness of the 0.5% formulation compared to Monistat 3^TM^ shows that occidiofungin is a promising candidate to be further tested as a VVC treatment.

Systemic fungal infections are more deadly than superficial infections with increased mortality and morbidity. Occidiofungin was evaluated for effectiveness in systemic candidemia [34]. Pharmacokinetic experiments showed intravenous administration of 2.5 mg/kg occidiofungin had peak plasma concentration of 364 ng/mL at 1 h post infection and a half-life of 3.2 h. IP administration had peak plasma concentration of 196 ng/mL at 24 h post infection and clearance within 48 h. Oral and subcutaneous administration had levels less than 12.5 ng/mL up to 48 h post administration indicating poor absorption through these routes. Drugs may be encapsulated by liposomes to improve their efficacy through improved pharmacokinetics, stability, or reduced toxicity. Amphotericin B is given in a liposomal formulation, where amphotericin B is integrated into the liposome bilayer, in order to reduce associated nephrotoxicity [35]. Intravenous administration of liposomal-encapsulated occidiofungin encapsulated by 1,2-dioleoyl-*sn*-glycero-3-phosphocholine (DOPC) and a 9:1 ratio of DOPC to 1,2-dipalmitoyl-*sn*-glycero-3-phospho-(1′-*rac*-glycerol) (DPPG) was evaluated. At 1 h post administration, 9:1 DOPC:DPPG liposomal occidiofungin had 10-fold higher levels of plasma concentration compared to liposome-free occidiofungin. Total clearance also extended from 18 h post administration to 48 h with liposomal occidiofungin. DOPC encapsulated occidiofungin had a four-fold increase in area under the concentration curve (AUC) and four-fold higher maximum plasma concentration (C_max_) but a shorter half-life of 0.4 h compared to occidiofungin. The 9:1 DOPC:DPPG liposomal occidiofungin had 17-fold higher AUC and 20-fold higher C_max_ as well as longer half-life by 3.83 h.

Give the superior pharmacokinetic properties of the DOPC:DPPG occidiofungin, this was chosen to be tested in a systemic infection model [34]. A 2.5 mg/kg dosage of DOPC:DPPG occidiofungin did not show significant reduction in fungal load in the kidneys of infected mice. However, a 5 mg/kg dosage of caspofungin showed a significant reduction of around 10-fold in fungal colony forming units. The pharmacokinetics of the DOPC:DPPG occidiofungin suggest there is sufficient levels of occidiofungin to exhibit antifungal activity. Both liposome-free and DOPC:DPPG occidiofungin had the same antifungal minimum inhibitory concentration values and similar kill kinetics in vitro, meaning that the liposomal formulation is not inhibiting antifungal activity.

The antifungal activity of occidiofungin and DOPC:DPPG occidiofungin was tested in vitro in the presence of serum and blood [34]. Drug binding to serum and blood proteins can have an inhibitory effect due to the inability to dissociate, reducing drug availability. Different species differ in their composition of blood and serum proteins, which could affect a drug’s activity. Sufficient levels of occidiofungin were detected in the blood of mice following organic solvent extraction; however, high serum protein binding is responsible for occidiofungin’s inability to treat the systemic infection. Both liposome-free and DOPC:DPPG occidiofungin showed an increase of more than 32-fold in MIC values in the presence of 50% mouse serum. Heat-inactivated mouse serum, or the addition of esterase inhibitors, still showed similar MIC values. Similar increase in MIC values is seen with rat serum, hamster blood, and guinea pig blood. The results suggest that rodent species may have serum protein compositions that are more prone to binding to occidiofungin. Porcine serum, beagle blood, and rhesus monkey blood showed a 32-fold increase in MIC values for DOPC:DPPG occidiofungin demonstrating lower protein binding but not at a significant level. Liposome-free occidiofungin had a 16-fold and 32-fold increase in MIC in the presence of human serum and blood respectively, and DOPC:DPPG occidiofungin had an 8- and 16-fold increase, respectively. Liposomally encapsulated occidiofungin formulation, in particular the DOPC:DPPG, could be used as an antifungal treatment due to improved pharmacokinetics compared to liposome-free occidiofungin whilst retaining in vitro antifungal activity. The results suggest that there is weaker binding of occidiofungin and DOPC:DPPG occidiofungin to human and primate serum proteins compared to other species. However, drug studies are always conducted on non-primate models prior to human trials, which is a concern for the development of occidiofungin to treat systemic infections. An occidiofungin analog which displays weaker binding to plasma proteins in mice or other animals may be developed to overcome this issue.

## 5. Summary

The lack of development of antifungal agents and continual rise of drug-resistant fungal infections highlights a need for a novel antifungal therapeutic option. The unique mechanism of action of occidiofungin makes it effective against a wide variety of fungal species, and resistance to occidiofungin has not yet been observed. Tolerance and lack of toxicity has been demonstrated in murine models. Occidiofungin is currently being developed for the treatment of VVC. Further investigation will be needed to determine whether occidiofungin could be used to treat fungal infections in other areas such as oral infection, skin infection, or systemic infection. An occidiofungin analog may be required to be developed to evaluate effectiveness in systemic infections, given the binding profile of occidiofungin to rodent plasma proteins. The development of occidiofungin may fulfill the desperate need for a new clinical antifungal agent.

## 6. Patents

Sano Chemicals Inc. has an exclusive license from Mississippi State University and Texas A&M University for the compound known as occidiofungin, as well as variants, applications, and methods of engineering. Several of these patents have been issued recently while several additional applications are pending with the USPTO as well as foreign patent offices. The patent strategy is based on securing the available rights in the compound and its production mechanism, also developing and securing rights in advancements to the compound while optimizing the production process to achieve better yields and decreased costs through proprietary manufacturing processes. Sano Chemicals along with its university partners monitor and enforce these patent rights to maintain a barrier of entry to competition. Securing and strengthening a robust intellectual property strategy is the foundation for promoting the continued develop of a first-in-class compound for the treatment of serious human diseases.

The current patents related to occidiofungin are listed below (Table 3). “Occidiofungin, a unique antifungal glycopeptide produced by a strain of *Burkholderia contaminans*” relates to the identification, production, and isolation of occidiofungin from *B. contaminans* MS14. In vitro antifungal activity data is also included with images showing disruption of fungal morphology when treated with occidiofungin. “Engineering the Production of a Conformational Variant of Occidiofungin That Has Enhanced Inhibitory Activity Against Candida Species” relates to compositions enriched for particular occidiofungin diastereomers/conformers and the methods and microorganisms for making compositions enriched for particular diastereomers/conformers. “Occidiofungin formulations and uses thereof” relates to occidiofungin formulations for the treatment of proliferative disorders, and the methods to make these formulations. “Synthesis of novel xylose free analogues and methods of using them” relates to methods of producing occidiofungin analogs lacking xylose, and the in vitro activity of these analogs.

## Figures and Tables

**Figure 1 antibiotics-11-01143-f001:**
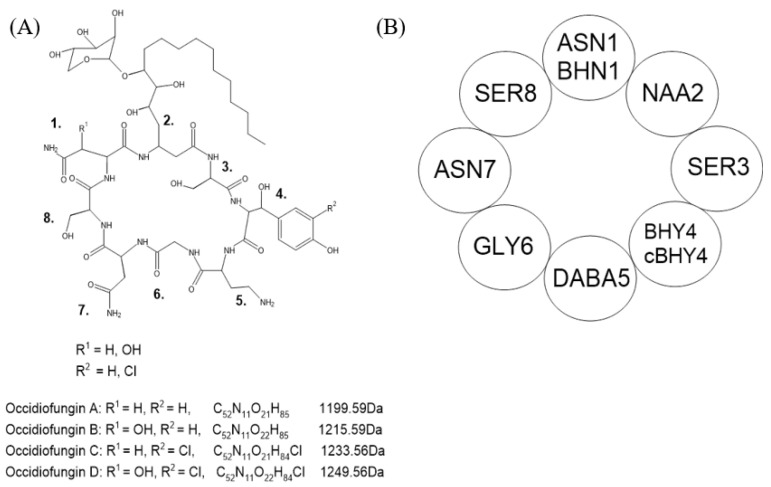
Chemical structure of occidiofungin. (A) Variants with their corresponding chemical formula and monoisotopic masses. Variants differ in R1 and R2 at amino acids designated as position 1 and 4. (B) Simplified structure with amino acids and their designated positions. Adapted from Gu et al. [12].

**Figure 2 antibiotics-11-01143-f002:**
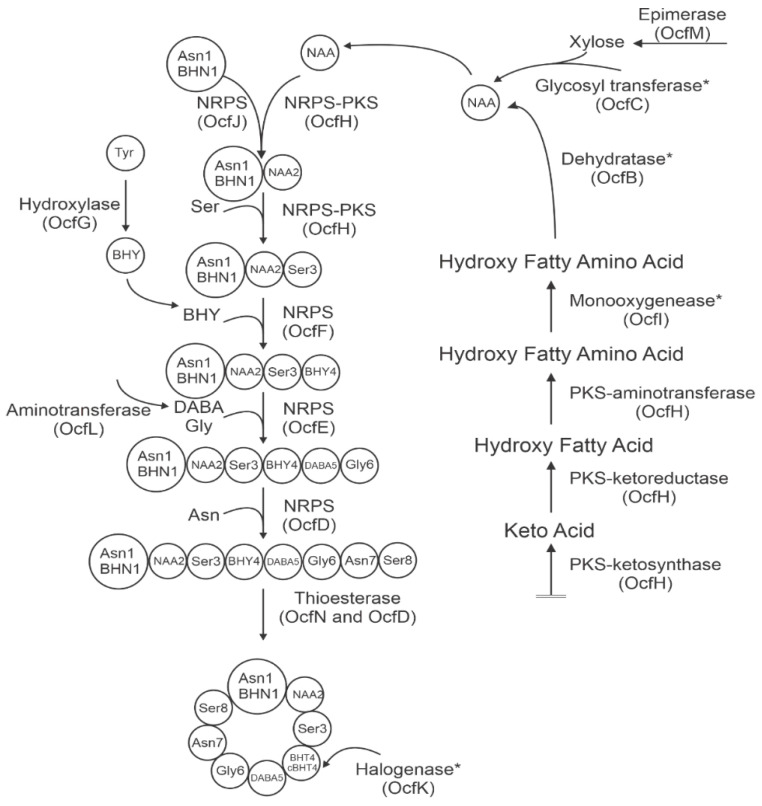
Biosynthetic pathway of occidiofungin consisting of a polyketide synthase pathway and non-ribosomally synthesized peptide synthase pathway. Steps which are uncertain in when they occur are marked with an asterisk. Adapted from Gu et al [12].

**Figure 3 antibiotics-11-01143-f003:**
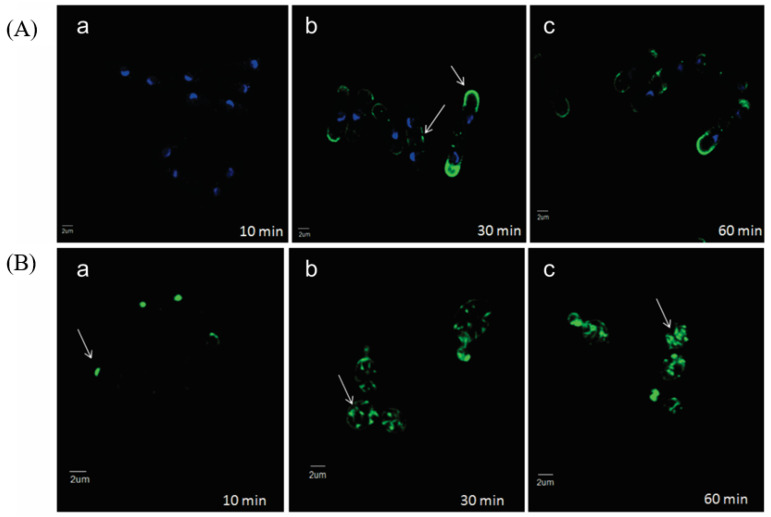
Localization of occidiofungin in yeast cells. (**A**) Alkyne derivatized occidiofungin is localized at the cell tips and division septa of dividing *Schizosaccharomyces pombe* cells by (b) 30 min and (60) min. (**B**) A time-course distribution of alkyne derivatized occidiofungin within *Saccharomyces cerevisiae* from 10 (a), 30 (b), and 60 (c) min. At 10 min, occidiofungin is localized at the bud tips of dividing *S. cerevisiae* cells. By 30 min, vesicular patterns are observed, which indicated endocytic vesicles coated by actin bound with occidiofungin. Localization of occidiofungin is demarcated with white arrows. Blue fluorescence is DAPI stain showing chromosome positions, and the green fluorescence is the azide-derivatized Alexa Fluor 488 stain bound to alkyne occidiofungin. Adapted from Ravichandran et al [9].

**Table 1 antibiotics-11-01143-t001:** Genes identified to be part of the *ocf* gene cluster. Gene homologs are compared to those found in *B. ambifara* AMMD. Predicted gene functions are based on *B. ambifara* AMMD and *B. lata* 383. Adapted from Gu et al [12].

Gene	Size (bp)	Homolog	Identity (%)	Predicted Function
*ambR1*	822	bamb_6466	89	LuxR-type regulator
*ambR2*	891	bamb_6468	77	LuxR-type regulator
*ocfA*	1704	bamb_6469	90	Cyclic peptide transporter
*ocfB*	483	bamb_6470	82	Hypothetic protein
*ocfC*	657	bamb_6471	94	Glycosyl transferase
*ocfD*	9495	bamb_6472	88	NRP synthetase
*ocfE*	9066	bamb_6473	89	NRP synthetase
*ocfF*	3921	bamb_6474	90	NRP synthetase
*ocfG*	1617	bamb_6475	93	Hydroxylase
*ocfH*	13,410	bamb_6476	91	Hybrid NRPS-PKS
*ocfI*	3324	bamb_6477	92	Flavin-dependent monooxygenase
*ocfJ*	4428	bamb_6478	91	NRP synthetase
*ocfK*	987	bamb_6479	91	Halogenase
*ocfL*	1371	bamb_6480	91	Transaminase
*ocfM*	951	bamb_6481	94	Epimerase
*ocfN*	720	bamb_6482	90	Thioesterase

**Table 2 antibiotics-11-01143-t002:** Minimum inhibitory concentration values of occidiofungin against various fugal species. Values provided are indicative of “100%” inhibition as dictated by National Committee for Clinical Laboratory Standards guidelines. Adapted from Ravichandran et al. [9].

Species	Occidiofungin (μg/mL)
*Rhizopus microsporus* 28506	8
*Rhizopus oryzae* 28403	8
*Rhizopus microsporus* 27785	4
*Mucor circinelloides* 19445	8
*Mucor racemosus* 27784	4
*Mucor fragilis* 27782	4
*Fusarium solani* 28386	4
*Rhizopus oryzae* 28403	4
*Fusarium oxysporum* 27718	4
*Fusarium solani* 18749	4
*Aspergillus flavus* 28517	4
*Aspergillus fumigatus* 28435	2
#*Candida albicans* 23512	1
*Candida albicans* 28200	8
*Candida albicans* 28102	2
#*Candida glabrata* 27243	4
*Candida glabrata* 25742	2
*Candida glabrata* 28271	8
*Candida krusei* 9541	4
#*Candida krusei* 28415	8
*Candida tropicalis* 9624	2
*Candida auris MRL*# 35646	0.25
*Candida auris MRL*# 35651	0.25

# Fungal species that exhibit fluconazole resistance.

**Table 3 antibiotics-11-01143-t003:** Patent overview.

Title	Patent Number	Patent Status
Occidiofungin, a unique antifungal glycopeptide produced by a strain of *Burkholderia contaminans*	US14/806,121—Continuation EP10780921.2A	Issued 9 October 2018 Issued 28 August 2019
Engineering the Production of a Conformational Variant of Occidiofungin That Has Enhanced Inhibitory Activity Against *Candida* Species	US17/113,764 EP2925774B1	Pending Issued on 31 January 2018
Occidiofungin formulations and uses thereof	US15/510,801 EP15840420.2	Granted 23 April 2019 Granted 3 March 2021
Synthesis of novel xylose free analogues and methods of using them	US17/045,283	Pending

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
