# Peer review of "Occidiofungin: Actin Binding as a Novel Mechanism of Action in an Antifungal Agent"

_antibiotics, 2022, doi:10.3390/antibiotics11091143_

Round 1

Reviewer 1 Report

Manuscript "Occidiofungin: Actin binding as a novel mechanism of action in an antifungal agent" presents an interesting summary of information on occidiofungin. Manuscript is well written, literature is well chosen, but newer articles need to be added.

Detailed comments:

Please provide the full names of the microorganisms in the abstract.

Table 2 - there are no references to the literature, some fungi repeat themselves with different MIC values (eg R. oryzae).

The authors may cite newer research results (only 9 out of 37 references are from the last 5 years).

Author Response

Please see the marked version of the manuscript to show the recommended changes.

Full names of the microorganisms are included in the abstract.

Comment: Table 2 - there are no references to the literature, some fungi repeat themselves with different MIC values (eg R. oryzae).

Reference to the MICs is in table legend and the table has been corrected to show the MIC values.

The authors may cite newer research results (only 9 out of 37 references are from the last 5 years).

The references were updated to reflect more recent publications. Many of the references are historical references on occidiofungin. (20/35) are now from the last five years.

Reviewer 2 Report

Dear Editor, 

Hansanant et.al. have presented a comprehensive review on the role of Occidiofungin , a glycolipopepdide as a novel antifungal agent and as a potential treatment for multidrug resistant microbes. The authors have elaborated on the synthesis, regulation, mechanism of action and biological activity of occidiofungin. Finally the authors conclude by highlighting the importance of developing antifungal agents which are less toxic and the need to develop occidiofungin analogs for treatment of multidrug resistant fungi. 

The authors have provided a comprehensive review and very detailed which is helpful to the reader. I just have a few minor suggestions that perhaps the authors could elaborate on. 

1. Is there any evidence of occidiofungin on the impact of biofilm formation since we know several fungi are capable of forming biofilms and therefore being more resistant to antimicrobials? 

2. Recently, a study was published in Phytopathology (Jia Jiayuan et al) where Occidiofungin ( from Burkholderia sp MS455) had antifungal effect on Aspergillus flavus. Since this is a comprehensive review it would be useful to include this alongwith the other Burkholderia sp ( Linbes 82-96)

Author Response

Please see the marked version of the manuscript to show the recommended changes.

Is there any evidence of occidiofungin on the impact of biofilm formation since we know several fungi are capable of forming biofilms and therefore being more resistant to antimicrobials?

Response: Some discussion to this question has been added to the review.  There have been no direct biofilm studies on the activity of occidiofungin.

2. Recently, a study was published in Phytopathology (Jia Jiayuan et al) where Occidiofungin ( from Burkholderia sp MS455) had antifungal effect on Aspergillus flavus. Since this is a comprehensive review it would be useful to include this alongwith the other Burkholderia sp ( Linbes 82-96)

Response: the suggested citation has been included.